

# Association between the expression difference of cortisol in umbilical cord blood and discordant growth in dichorionic twins: a cross-sectional survey

Yimin Huang[1,*], Hui Zhu[1,*], Yi Li[2], Jianguo Wang[1] and Li Ni[1]

[1] Jiaxing Maternity and Child Health Care Hospital, Affiliated Women and Children Hospital of Jiaxing University, Jiaxing, China

[2] Zhejiang Chinese Medical University, Hangzhou, Zhejiang, China

[*] These authors contributed equally to this work.

## ABSTRACT

**Background**. This study aimed to assess whether the expression difference of cortisol in umbilical cord blood is associated with dichorionic twin inconsistent growth.

**Methods**. This research included 108 patients with dichorionic twin pregnancy delivered at Jiaxing Maternity and Child Health Care Hospital between January 2021 and December 2024. Depending on whether or not they had twin inconsistent growth, participants were classified into the discordant twins (DT) group (47 pairs of discordant twins) as the experimental group and the concordant twins (CT) group (61 pairs of concordant twins) as the control group. According to the medical record, the maternal and neonatal information of the two groups were collected, and the differences in the basic conditions between the two groups of maternal and twin fetuses were analyzed. The fetal birth weight of the two groups was weighed, and the cortisol content of the umbilical cord blood was measured by chemiluminescence.

**Results**. There were no significant differences in maternal age, gestational age (GA), parity, body and mass index (BMI), mode of conception, delivery time, and fetal gender between the two groups of maternal characteristics. There were also no significant differences in the amniotic fluid depth, umbilical artery standard deviation (SD) value, umbilical cord entanglement, placental shape, and twin gender differences between the two groups of neonates. However, the smaller neonates in the DT group had a higher cortisol level in the umbilical cord blood, with a significant difference, while there was no significant difference in cortisol levels in the umbilical cord blood of the CT group of twin fetuses.

**Conclusions**. The difference in cortisol levels in the umbilical cord blood of twin fetuses may be related to the occurrence of inconsistent growth of dichorionic twins.

Corresponding author
Li Ni, 530542192@qq.com

## INTRODUCTION

Twin growth inconsistency is defined as significant differences in birth weight between twin fetuses, and its diagnostic criteria is still controversial (ranging from 15 to 25%) (*Miller, Chauhan & Abuhamad, 2012*). With the maturity of assisted reproductive technology (ART) in China, the incidence of twin discordance is increasing, which is one of the causes of perinatal fetal death (*D'Antonio et al., 2018*; *Park et al., 2021*; *Rissanen et al., 2022*). Fetal intrauterine growth is jointly regulated by maternal, environmental, and fetal factors (*Anonymous, 2021*; *Sagi-Dain, 2021*). Some previous studies have suggested that the birth weight of fetuses may be related to the various hormone levels exposed during pregnancy (*Canpolat et al., 2011*). However, the causes of twin discordance are still unclear (*Kadioglu Simsek et al., 2019*). Cortisol, as a glucocorticoid, has been reported to affect fetal birth weight (*Yu et al., 2022*; *Thompson et al., 2024*). Literature research suggested that excessive exposure to corticosteroids during pregnancy can lead to lower birth weight (*Zhu et al., 2019*), but the effect of cortisol levels on twin discordance has not been reported.

In this study, dichorionic discordant twins were selected as the research object. The content of cortisol in fetal umbilical cord blood was determined by chemiluminescence to investigate the relationship between the expression difference of cortisol in fetal umbilical cord blood and the inconsistent growth of twins, in order to provide better prevention and treatment basis.

## MATERIAL AND METHODS

### Group setting and data collection

According to the inclusion and exclusion criteria listed below, dichorionic twins delivered at Jiaxing Maternity and Child Health Care Hospital from January 2021 to December 2024 were selected. Among them, those diagnosed with discordant growth were admitted to the discordant twins (DT) group, while others without discordant growth were admitted to the concordant twins (CT) group.

This research was approved by the Medical Ethics Committee of Jiaxing Maternity and Child Health Care Hospital (No. 2020-1). All participants signed the informed consent form after reading the information sheet and an oral project explanation. The research complies with the Helsinki Declaration of Human Rights (revised in Seoul in 2008).

Inclusion criteria: ① dichorionic twins with no fetal malformations, and the inter twin birth weight discordance difference of 25% or more according to the recent Delphi consensus (*Khalil et al., 2019*); ② fetal umbilical vein blood was collected and detected by chemiluminescence after caesarean section; ③ informed consent from patients; ④ complete clinical case data.

Exclusion criteria: ① fetal congenital malformations; ② prenatal corticosteroids therapy; ③ maternal presence of gestational diabetes, gestational hypertension or other endocrine diseases such as hyperthyroidism, hypothyroidism, polycystic ovary syndrome during pregnancy.

Maternal and neonatal characteristics were collected from the medical record database, the maternal information including maternal age, gestational age (GA), parity, maternal

body and mass index (BMI), method of conception, delivery time, and gender of twins; the neonatal information including amniotic fluid depth during pregnancy, SD value of umbilical artery, umbilical cord wrapped around neck, placenta shape, and fetal birth weight. The relevant diagnostic criteria refer to the selective fetal growth restriction (sFGR) definition for dichorionic twin pregnancies in the Delphi consensus 2019 (*Khalil et al., 2019*). Fetuses were weighed after delivering on the day of caesarean section. After placenta expulsion, two mL of umbilical cord blood was collected, and serum was taken after centrifugation at 3,500 rpm for 5 min. The content of cortisol in umbilical cord blood was determined by chemiluminescence.

### Detection of cortisol level in umbilical cord blood

For the measurement of cortisol, umbilical cord blood was treated according to manufacturer of the Cortisol *in vitro* diagnostic kits (chemiluminescent immunoassay) (Mindray, China, Shenzhen). Briefly, the sample was added to the reaction tube with superparamagnetic particles (magnetic beads) coated with goat anti-rabbit IgG, polyclonal rabbit anti-cortisol antibody and cortisol-alkaline phosphatase conjugate. After the reaction was completed, the chemiluminescent substrate (3-[2-spiroadamatane]-4-methoxy-4-[3-phosphoryloxy]-phenyl-1,2-dioxetane Dioxetane, AMPPD) was added to the reaction tube. Subsequently, cortisol levels were measured by using the Mindray CL-6000i Clinical Fully Automated Chemiluminescence Immunoassay System Analyzer.

### Statistical analysis

All the data were analyzed by SPSS software (SPSS Inc., Chicago, IL) version 27.0. Categorical variables were presented as number (percentage) and were compared using the Pearson chi-square test or Fisher's exact test. Normally distributed measurement data were expressed as mean $\pm$ standard deviation ($\bar{x} \pm s$), and the comparison between the two groups was performed using the $t$-test. Non-normally distributed data were expressed as median with interquartile range *[M (P25, P75)]* and analyzed using the Mann–Whitney U test or the Kruskal-Wallis test. A *P*-value of < 0.05 was considered statistically significant.

## RESULTS

### Comparison of maternal and neonatal characteristics between the two groups

A total of 108 cases of dichorionic twin pregnancies who met the selection criteria were enrolled in this study (Fig. 1). Of them, 47 cases were admitted to the DT group and 61 cases were admitted to the CT group. The maternal baseline characteristics of the two groups were summarized in Table 1, and there was no statistical significance in the difference (*P* > 0.05).

The comparison of neonatal baseline characteristics between larger and smaller neonates in the DT group is shown in Table 2. Each neonatal twin in the CT group was further randomly divided into twin A and twin B due to the similar birth weights. There were no significant difference in amniotic fluid depth, umbilical artery SD value, umbilical cord around the neck, placental shape, and twin sex between the two groups (*P* > 0.05), as is shown in Tables 2 and 3.
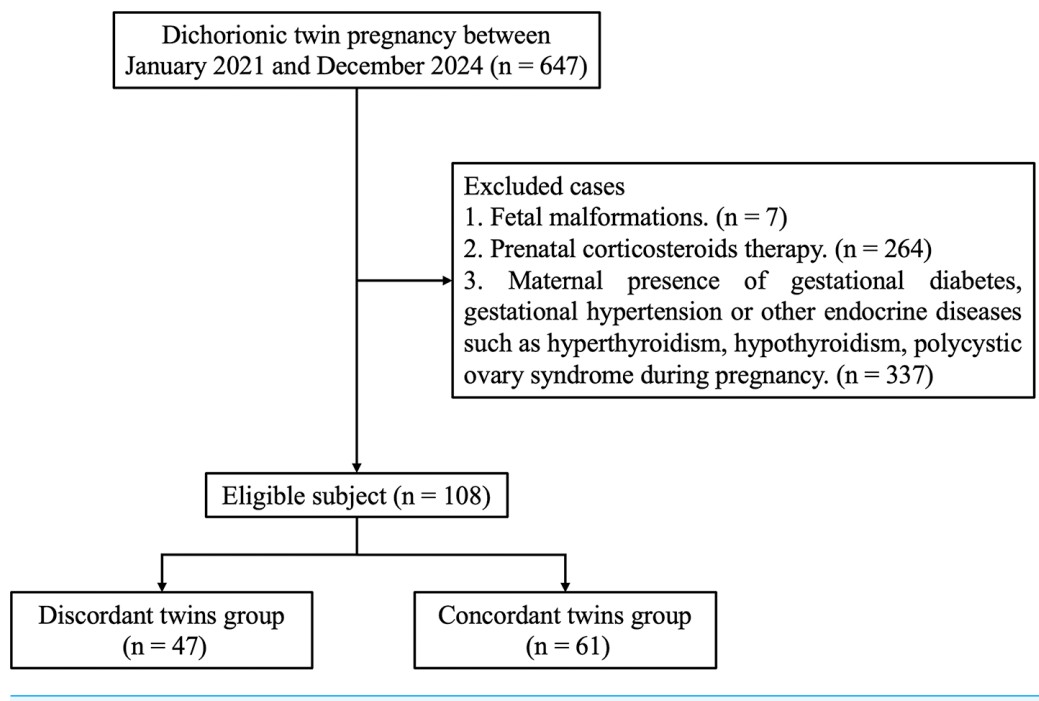

**Figure 1 Flowchart of study inclusion.**

## Comparison of cortisol levels in umbilical cord blood between two groups of neonates

As is shown in Tables 4 and 5, the difference of cortisol levels between larger and smaller neonates in the DT group was statistically significant ($P < 0.05$), while there was no statistical significant difference in cortisol levels between the CT group neonates ($P > 0.05$). Further study found that there were significant differences in cortisol levels between the CT group and the DT group, and the cortisol levels of the other three subgroups were significantly lower than those of the smaller neonates in the DT group (Table 6).

## DISCUSSION

Discordant twin growth is associated with many risk factors, including advanced maternal age, the use of assisted reproductive technology, gestational hypertension, and discordant fetal sex (*Qiao et al., 2019*). In recent years, reported research have focused on the effect of placental function on the development of twins. It is believed that placental villus dysplasia or small mass of placenta during pregnancy may lead to inconsistent growth of twins (*Kent et al., 2012*; *Guo, Sun & Yang, 2017*). In this study, chi-square tests were performed on the above known risk factors, and no significant differences were found between the two groups. Therefore, the focus was shifted to the relationship between the placental function and discordant twin growth.

Previous studies have shown that the level of glucocorticoids in fetal cord blood is closely related to the birth weight of the fetus (*Jin et al., 2019*; *Schneider et al., 2021*). Cortisol, as a type of glucocorticoid, is a product of the hypothalamus-pituitary-adrenal

**Table 1  Comparison of maternal characteristics between the two groups.**

| Baseline characteristics | DT group ($N = 47$) | CT group ($N = 61$) | $X^2$ | $P$ |
|---|---|---|---|---|
| Maternal age (year) | | | 1.792 | 0.181 |
|    >30 | 18 (38.30) | 16 (26.23) | | |
|    ≤30 | 29 (61.70) | 45 (73.77) | | |
| GA at birth (week) | | | 0.907 | 0.341 |
|    >34 | 47 (100.00) | 61 (100.00) | | |
|    ≤34 | 0 (0.00) | 0 (0.00) | | |
| Parity | | | 0.418 | 0.518 |
|    <2 times | 40 (85.11) | 49 (80.33) | | |
|    ≥2 times | 7 (14.89) | 12 (19.67) | | |
| Maternal BMI | | | 0.067 | 0.795 |
|    >25 kg/m² | 36 (76.60) | 48 (78.69) | | |
|    ≤25 kg/m² | 11 (23.40) | 13 (21.31) | | |
| ART used | | | 0.276 | 0.600 |
|    Yes | 36 (76.60) | 44 (72.13) | | |
|    No | 11 (23.40) | 17 (27.87) | | |
| Delivery time | | | | |
| 0:01–10:00 | 36 (76.60) | 47 (77.05) | 0.798 | 0.705 |
| 10:01–18:00 | 7 (14.89) | 8 (13.11) | | |
| 18:01–24:00 | 4 (8.51) | 6 (9.84) | | |
| Consistent twin fetal sex | | | 0.009 | 0.925 |
|    Yes | 22 (46.81) | 28 (45.90) | | |
|    No | 25 (53.19) | 33 (54.10) | | |

**Notes.**

GA, gestational age; BMI, body and mass index; ART, assisted reproductive technology

axis, which is released to regulate various important functions and maintain metabolic balance in the body during stress responses. It has been shown that cortisol plays an important role in fetal growth and development (*Konstantakou et al., 2017*). The minority of fetal cortisol is secreted by the fetal hypothalamic-pituitary-adrenal (HPA) axis during late pregnancy, whereas the majority is transferred from the mother *via* placental blood circulation. The two isozymes of 11β-hydroxysteroid dehydrogenase (11β-HSD) serve as the natural placental glucocorticoid barrier, regulate the level of cortisol transferred from the mother to the fetus (*Zhao et al., 2024*). 11β-HSD1 catalyzes the regeneration of active glucocorticoids, 11β-HSD2 makes glucocorticoids inactivating (*Chapman, Holmes & Seckl, 2013*). In dichorionic twin pregnancies, each fetus has its own placenta, allowing for distinct placental barrier functions due to independent genetic makeup. It can result in variability in how each placenta manages the selective transport of nutrients and waste, which potentially leads to the discordant in fetal growth and development (*Park et al., 2021*). In contrast, monochorionic twins share a single placenta and possess identical genetic backgrounds, which leads to similar placental barrier functions (*Groene et al., 2022*). This shared placental structure and function in monochorionic twins makes it challenging to directly compare and differentiate the placental barrier effects between the two fetuses.

**Table 2  Comparison of neonatal baseline characteristics between larger and smaller neonates in DT group.**

| Baseline characteristics | Larger neonates (N = 47) | Smaller neonates (N = 47) | $X^2$ | P |
|---|---|---|---|---|
| Normal amniotic fluid depth | | | 0.646 | 0.421 |
| Yes | 40 (85.11) | 37 (78.72) | | |
| No | 7 (14.89) | 10 (21.28) | | |
| Normal umbilical artery SD value during pregnancy | | | 1.099 | 0.294 |
| Yes | 36 (76.60) | 40 (85.11) | | |
| No | 11 (23.40) | 7 (14.89) | | |
| Umbilical cord around neck | | | 1.016 | 0.313 |
| Yes | 8 (17.02) | 12 (25.53) | | |
| No | 39 (82.98) | 35 (74.47) | | |
| Normal placental shape | | | 0.213 | 0.646 |
| Yes | 45 (95.74) | 44 (93.62) | | |
| No | 2 (4.26) | 3 (6.38) | | |
| Consistent twin fetal sex | | | 0.000 | 1.000 |
| Yes | 22 (46.81) | 22 (46.81) | | |
| No | 25 (53.19) | 25 (53.19) | | |

**Table 3  Comparison of neonatal baseline characteristics between twin A and twin B in CT group.**

| Baseline characteristics | Twin A (N = 61) | Twin B (N = 61) | $X^2$ | P |
|---|---|---|---|---|
| Normal amniotic fluid depth | | | 0.775 | 0.379 |
| Yes | 53 (86.89) | 56 (91.80) | | |
| No | 8 (13.11) | 5 (8.20) | | |
| Normal umbilical artery SD value during pregnancy | | | 0.436 | 0.509 |
| Yes | 55 (90.16) | 57 (93.44) | | |
| No | 6 (9.84) | 4 (6.56) | | |
| Umbilical cord around neck | | | 0.518 | 0.472 |
| Yes | 9 (14.75) | 12 (19.67) | | |
| No | 52 (85.25) | 49 (80.33) | | |
| Normal placental shape | | | 0.701 | 0.402 |
| Yes | 57 (93.44) | 59 (96.72) | | |
| No | 4 (6.56) | 2 (3.28) | | |
| Consistent twin fetal sex | | | 0.000 | 1.000 |
| Yes | 28 (45.90) | 28 (45.90) | | |
| No | 33 (54.10) | 33 (54.10) | | |

In this study, after excluding the differences in basic maternal and neonatal characteristics between the two groups, chemiluminescence was used to detect the cortisol levels in the umbilical cord blood of neonates in both groups. The results showed that the cortisol levels of the smaller neonates in the DT group were significantly higher compared to the

**Table 4 Comparison of cortisol levels in umbilical cord blood between larger and smaller neonates in the DT group.**

|  | N | Birth weight (g) [M (P25, P75)] | Cortisol (nmol/L) [M (P25, P75)] |
|---|---|---|---|
| Larger neonates | 47 | 2,435.00 (2,386.00 ~2,498.00) | 98.00 (94.00 ~105.00) |
| Smaller neonates | 47 | 1,918.00 (1,886.00 ~1,989.00) | 233.00 (198.00 ~247.00) |
| z |  | −8.201 | −7.870 |
| P |  | <0.001 | <0.001 |

**Table 5 Comparison of cortisol levels in umbilical cord blood between twins in the CT group.**

|  | N | Birth weight (g) [M (P25, P75)] | Cortisol (nmol/L) [M (P25, P75)] |
|---|---|---|---|
| Twin A | 61 | 2,418.00 (2,379.50, 2,454.00) | 102.00 (97.00 ~106.00) |
| Twin B | 61 | 2,433.00 (2,398.50, 2,472.00) | 99.00 (92.75 ~105.50) |
| z |  | −1.700 | −1.364 |
| P |  | 0.089 | 0.173 |

**Table 6 Comparison of cortisol levels in umbilical cord blood between the DT and CT groups.**

|  | Birth weight (g) [M (P25, P75)] | Cortisol (nmol/L) [M (P25, P75)] |
|---|---|---|
| Smaller neonates (N = 47) | 1,918.00 (1,886.00, 1,989.00) | 233.00 (198.00, 247.00) |
| Larger neonates (N = 47) | 2,435.00 (2,386.00, 2,498.00)[a] | 98.00 (94.00, 105.00)[a] |
| Twin A (N = 61) | 2,418.00 (2,379.50, 2,454.00)[b] | 102.00 (97.00, 106.00)[b] |
| Twin B (N = 61) | 2,433.00 (2,398.50, 2,472.00)[c] | 99.00 (92.75, 105.50)[c] |
| H | 110.677 | 99.864 |
| P | < 0.001 | < 0.001 |

Notes.
[a] There are statistical differences between larger neonates and smaller neonates.
[b] There is a statistical difference between twin A and smaller neonates.
[c] There are statistical differences between twin B and smaller neonates.
All comparisons were adjusted using the Bonferroni correction.

larger twin neonates ($P < 0.05$). However, there was no statistically significant difference in the cortisol levels in the umbilical cord blood of neonates in the CT group ($P > 0.05$). The impaired placental glucocorticoid regulation has been reported as one of the causes of fetal intrauterine growth restriction (IUGR) in singleton fetuses, which leads excessive glucocorticoids to the fetus (*Kent et al., 2012*; *Jin et al., 2019*). Moreover, the growth restriction pattern of monochorionic discordant twin fetuses was the same as that in singleton fetuses with IUGR (*Park et al., 2021*; *Coutinho Nunes et al., 2016*). IUGR has been associated with the reduced activity of the enzyme 11β-HSD2 (*Chatuphonprasert, Jarukamjorn & Ellinger, 2018*). In recent studies, the expression levels of 11β-HSD1 mRNA in periumbilical subcutaneous adipose tissue of adult male twins have been investigated. The results indicate a significant elevation of 11β-HSD1 mRNA expression in heavier twins, with statistical significance observed ($P < 0.05$) (*Vihma et al., 2017*). However, this study did not analyze the expression of 11β-HSD2, as it is not expressed in adipose tissues (*Kupczyk et al., 2022*). These findings suggest a correlation between the differential expression of 11β-HSD1 and variations in twin body mass. 11β-HSD1 serves as a key

enzyme in regulating glucocorticoid activity within adipose tissues, influencing metabolic processes and body weight. Therefore, it is considered that the occurrence of fetal growth discordance in the DT group may be related to excessive glucocorticoids entering the fetus through the placental barrier.

Corticosteroids contribute to embryo implantation in early pregnancy, promote fetal adrenal development between the 7th and 14th weeks of pregnancy, inhibit dehydroepiandrosterone (DHEA) synthesis and promote female genital development in the last three months of pregnancy (*Agnew et al., 2018*; *Zheng et al., 2020*; *Asztalos, Murphy & Matthews, 2022*). Prior to birth, fetal serum corticosteroid levels increase significantly to ensure normal development of the fetal lungs and several other organs (*Busada & Cidlowski, 2017*). Insufficient transmission of corticosteroids from the mother to the fetus during pregnancy can impair fetal lung function and increase neonatal mortality, while excessive transmission can suppress fetal growth and development and increase the risk of cardiovascular and metabolic diseases such as hypertension, impaired glucose tolerance, diabetes, and stroke in adulthood (*Chatuphonprasert, Jarukamjorn & Ellinger, 2018*).

## CONCLUSIONS

In conclusion, after excluding the influence of relevant risk factors, the cortisol levels in the smaller neonate of dichorionic discordant twins was significantly higher than that in the larger neonate, which may be related to the occurrence of discordant growth in twins.

Up to now, research on the relationship between the differences in cortisol levels in umbilical blood and discordant growth in twins has been relatively limited. Growth and development tracking is needed in the future, and the amount of supplementation needs to be further studied. In addition, the regulatory mechanisms of placental 11β-HSD and its specific associations with growth discordance in twins remain to be explored. These research findings can deepen our understanding of the mechanisms underlying growth discordance in twins and provide new insights and strategies for the clinical management of twin pregnancies.

### Funding
This work was supported by Zhejiang Provincial Medical and Health Science and Technology Project (2021KY1121), Jiaxing Municipal Public Welfare Research Plan (2021AD30130). Natural Science Foundation of Zhejiang Province (LQ21H160040) and Jiaxing Science and Technology Bureau (LGF20H090020). The funders had no role in study design, data collection and analysis, decision to publish, or preparation of the manuscript.

### Grant Disclosures
The following grant information was disclosed by the authors:
Zhejiang Provincial Medical and Health Science and Technology Project: 2021KY1121.
Jiaxing Municipal Public Welfare Research Plan: 2021AD30130.

Natural Science Foundation of Zhejiang Province: LQ21H160040.
Jiaxing Science and Technology Bureau: LGF20H090020.

## Competing Interests

The authors declare there are no competing interests.

## Author Contributions

- Yimin Huang analyzed the data, authored or reviewed drafts of the article, and approved the final draft.
- Hui Zhu performed the experiments, authored or reviewed drafts of the article, and approved the final draft.
- Yi Li performed the experiments, prepared figures and/or tables, and approved the final draft.
- Jianguo Wang analyzed the data, prepared figures and/or tables, authored or reviewed drafts of the article, and approved the final draft.
- Li Ni conceived and designed the experiments, authored or reviewed drafts of the article, and approved the final draft.

## Human Ethics

The following information was supplied relating to ethical approvals (i.e., approving body and any reference numbers):

This study was approved by the Medical Ethics Committee of Jiaxing Maternity and Child Health Care Hospital (No. 2020-1).

## Data Availability

The raw data is available in the Supplemental File.

## Supplemental Information

Supplemental information for this article can be found online at http://dx.doi.org/10.7717/peerj.19479#supplemental-information.

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
