# Peer review of "Association between the expression difference of cortisol in umbilical cord blood and discordant growth in dichorionic twins: a cross-sectional survey"

_PeerJ, doi:10.7717/peerj.19479_

## Round 0.1 · original submission · Major Revisions

Several issues to be clarified before publication. Please address the concerns of the reviewers in detail.

Reviewer 1 ·

Basic reporting

Clear, unambiguous, professional English language used throughout. Structure is clarity.

Experimental design

Scientifically.

Validity of the findings

a. Line 63 “the inter twin birth weight discordance difference of 15 % or more”. Why you choose this criterion?
b. The present study included a relatively small number of patients, fewer variables, and simple conclusion, which make the the article is not very readable.
c. The part of conclusion was too simple. What is the relationship between cortisol in umbilical cord blood, placenta, and mother? what are the possible mechanisms, and why DCDA was chosen as the research object instead of MCDA or singleton pregnancy? Please enrich your discussion section.

Additional comments

none

Reviewer 2 ·

Basic reporting

This is a good study with some implications for clinical practice. However, there are still some issues that need further discussion.

Experimental design

The authors are advised to update the grouping criteria based on the sFGR definition for dichorionic twin pregnancies in the Delphi consensus 2019 (PMID: 29363848) to align with international standards.
For the definition of inconsistent growth of DCDA twins:
one solitary parameter: estimated fetal weight (EFW) of one twin < 3rd centile.
at least two out of four contributory parameters: (1) EFW of one twin < 10th centile, (2) EFW discordance of ≥ 25%, (3) umbilical artery pulsatility index of the smaller twin > 95th centile.

Validity of the findings

In line 116-120, the author mentioned reported researches found unequal placental may lead to inconsistent growth of twins. The results of this study show no difference in placental shape between the two groups. What’s the reason?

Additional comments

In lines 135-136, please provide additional references to analyze the relationship between excessive glucocorticoids and fetal growth discordance.

---

## Round 0.2 · Major Revisions

The submission still needs some clarifications.

Reviewer 1 ·

Basic reporting

no comment

Experimental design

Please provide additional details on how the cortisol levels were measured.

Validity of the findings

a. The flowchart reported 486 cases of dichorionic twin pregnancies from 2021 to 2023, with some delivering before 34 weeks of gestation, but no cases received corticosteroid therapy before pregnancy. And Table 1 indicated that out of 84 twin pregnancies, only 17 were consistent twin fetal sex. These findings are not consistent with clinical practice. Please verify the data.
b. The findings of higher cortisol levels in smaller neonates of the DT group and the lack of significant differences in the CT group are intriguing and contribute to the understanding of inconsistent growth of dichorionic twins. But I suggest you to compare the cortisol levels in umbilical cord blood between the DT and CT groups.

Additional comments

a. Mericq et al. (Eur J Endocrinol, 2009 Sep;161(3):419-25) have shown that the activity of 11beta-HSDs in placentas may be related to fetal sex. Please explain whether there are any differences in fetal sex among the groups studied and consider if these differences could potentially influence the results.
b. Moreover, please consider including suggestions for future research directions, such as the long-term impacts of cortisol levels on twin health.

---

## Round 0.3 · accepted · Accept

I feel it is well revised.